# Is Fear of Harm (FoH) in Sports-Related Activities a Latent Trait? The Item Response Model Applied to the Photographic Series of Sports Activities for Anterior Cruciate Ligament Rupture (PHOSA-ACLR)

**DOI:** 10.3390/ijerph17186764

**Published:** 2020-09-16

**Authors:** Wim van Lankveld, Ron J. Pat-El, Nicky van Melick, Robert van Cingel, J. Bart Staal

**Affiliations:** 1Musculoskeletal Rehabilitation Research Group, HAN University of Applied Sciences, 6503 GL Nijmegen, The Netherlands; bart.staal@han.nl; 2Department of Methods and Statistics, Open University of The Netherlands, 6401 DL Heerlen, The Netherlands; ron.pat-el@ou.nl; 3Knee Expert Center Company Eindhoven, 5624 EB Eindhoven, The Netherlands; nicky@knieexpertisecentrum.nl; 4Radboud Institute for Health Sciences, IQ Healthcare, University Medical Center, 6500 HB Nijmegen, The Netherlands; r.vancingel@smcp.nl; 5Sport Medisch Centrum Company Papendal, 6816 VD Arnhem, The Netherlands

**Keywords:** anterior cruciate ligament reconstruction (ACLR), kinesiophobia, fear of harm/movement/injury, validation

## Abstract

*Background*: Fear of Harm (FoH) predicts return to sports in Anterior Cruciate Ligament Reconstruction (ACLR) and can be assessed using the Photographic Sports Activities for ACLR (PHOSA-ACLR). This study was conducted to determine whether FoH assessed using the PHOSA-ACLR is a latent trait, and to analyze differences in PHOSA-ACLR in athletes with or without an ACL rupture. *Methods*: Three convenience samples completed the PHOSA-ACLR: (1) ACLR patients (*n* = 58; mean age 25.9 years; range 17–56; SD = 8.2; 43% male); (2) first year Physical Therapy (PT) students (*n* = 169; mean age = 19.2; SD = 2.0; 48% male), and (3) junior football players (*n* = 30; mean age = 18.3; range 17–20; SD = 3.2; 94% males). ACLR patients additionally reported functioning and Fear of Movement. PHOSA-ACLR items were analyzed with Item Response Theory using the Graded Response Model (GRM). Differences between three groups of participants were analyzed using Univariate Analysis of Variance. *Results*: Data fitted the two-parameter GRM, and therefore the items of the PHOSA-ACLR constitute a latent trait. There was a significant difference between the three groups in PHOSA-ACLR after controlling for age and gender (F (2, 255) = 17.1, *p* < 0.001). PT students reported higher levels of FoH compared to either ACLR patients or healthy soccer players. *Conclusions:* PHOSA-ACLR items constitute a latent trait of FoH for ACLR-specific movements. Contrary to expectations, PHOSA-ACLR is higher in first year physiotherapy students compared to patients rehabilitating from ACLR, and healthy junior soccer players.

## 1. Introduction

An Anterior Cruciate Ligament (ACL) tear is a frequently reported injury in sports activities that include pivoting movements of the knees, such as soccer, basketball, football, handball, and skiing. The incidence of ACL tears is estimated at 68.6 per 100,000 person every year [1]. Most patients recover to normal function within 6–8 weeks [2], but only half of the patients return to pre-injury sports levels [3,4]. Fear of movement or (re)injury is an important factor in rehabilitation after sports-related injury [5,6]. There are several pathways by which such Fear of Harm (FoH) can have an impact on recovery after a sports-related injury, such as Anterior Cruciate Ligament Reconstruction (ACLR) [5]. Physiological arousal associated with fear can induce increased muscle tension, fatigue, decreased coordination, and muscular guarding thus escalating the individual’s susceptibility to injury [7]. Trigsted et al. [8] showed that fear of re-injury is associated with stiffened movement patterns associated with increased risk of a second ACL injury. The fear avoidance model of pain offers another pathway in which FoH can hinder recovery [9]. The model holds that adverse events might trigger negative emotions and cognitions that can become conditioned. In this way FoH can become internalized, preventing a person’s engagement in activities necessary for recovery [9]. In ACLR rehabilitation, FoH has a negative impact on return to sports after ACL rupture and reconstruction [10,11]. However, intensity of these fears differs between individuals from non-existent to maladaptive psychological outcomes and disorders (anxiety disorders), as in kinesiophobia [12,13]. Therefore, it is important to assess the individual patient’s level of FoH after ACLR. 

In those ACLR patients with disruptive levels of FoH, addressing task-specific FoH in rehabilitation might help patients return to their pre-injury sport or recreational activity [14,15]. Fear can be treated using exposure, which has been successfully applied in chronic low-back pain, complex regional pain syndrome and other conditions [16]. Graded exposure guides the patient through a hierarchical series of fear eliciting movements and activities that have been avoided. McArdle described a cognitive-behavioral approach based on exposure to treat dysfunctional FoH in one person with ACLR [17]. However, results of exposure treatment have not been reported in the ACLR rehabilitation literature. One reason might be the lack of instruments to determine a hierarchy of perceived harmfulness of physical activities and movements specific for ACLR. Fear in ACLR is most frequently assessed using generic measures of Fear of Movement, such as the Tampa Scale of Kinesiophobia (TSK) [18], or the ACL-Return to Sports Injury Scale (ACL-RSI) [19]. However, both scales do not provide information about specific FoH induced by specific sports-related tasks in the individual with ACLR, preventing return to sports. Therefore, the Photographic Sports Activity-Anterior Cruciate Ligament Reconstruction (PHOSA-ACLR) was developed to assess fear for specific sports-related activities [20]. 

The PHOSA-ACLR gives specific information about fear invoking sports situations that is not measured by other measures. Traditional psychometric properties, such as construct validity, internal consistency and test–retest reliability, underlined the PHOSA-ACLR’s validity and reliability [20]. However, an important limitation of these traditional psychometrics is the assumption that measurement precision is constant across the entire trait range, or that each item in a test contributes equally to the test score [21]. As a result, measurement precision may not be constant for all respondents, as is often true in practice. Especially tests used for clinical purposes are good at differentiating among people in the high range of a trait, but less well-suited for differentiating people in the normal-to-low range of the trait, which is where the at-risk patients usually reside. For clinical use, it is important to know how the individual items on the PHOSA-ACLR are related to each other hierarchically based on respondents’ responses. Based on such a fear hierarchy of individual items, clinicians can use the instrument in exposure treatment for these patients. To establish such a hierarchy of items, latent trait analysis using Item Response Theory (IRT) can be used. IRT allows for the identification of items that are indicative of high ability, and potentially how well these items discriminate around that ability. The better the item discriminates, the stronger its diagnostic information, and thus value, of the item in estimating FoH in clinical practice.

Therefore, this study was conducted to determine whether FoH assessed using the PHOSA-ACLR is a latent trait using Item Response Theory Models. Furthermore, differences in PHOSA-ACLR scores between patients with a recent ACLR and healthy athletes were determined. It is expected that patients with ACLR will have higher scores on FoH assessed with the PHOSA-ACLR compared to healthy controls.

## 2. Materials and Methods

### 2.1. Participants

For this study, we used cross-sectional data from three convenience samples of subjects. The *first* sample consisted of 58 ACLR patients, who participated in a study conducted in 2015 to measure psychometric dimensions relevant to FoH after ACLR [20]. Patients were included from Physical Therapy (PT) practices when they were between 3 months and 3 years after their ACLR. The *second* sample of respondents consisted of 169 healthy regular students at the Hogeschool Arnhem Nijmegen (HAN) University of Applied Sciences. All first-year PT students attending a lecture in the first month of their PT education were asked to complete the PHOSA-ACLR. At the start of the lecture, students were informed about the study, and asked to provide written informed consent. The *third* group of participants consisted of 30 healthy junior soccer players that were enrolled in a training program of the professional Soccer club Vitesse (Arnhem), additional to their regular educational program. The study including patients was approved by the Ethical Advisory Board, Faculty of Health, Behavior and Society of the HAN University of Applied Sciences. Committee reference number is ACPO 21.03/16. The study including students was approved by the same Ethical Advisory Board; Committee reference number EACO 71.05/17.

### 2.2. Measurements

All participants reported demographics (gender, age), and completed the PHOSA-ACLR using an online questionnaire (sample 1) or printed questionnaire (sample 2 and 3). The PHOSA-ACLR is comprised of photographic images of 12 sports situations invoking FoH in ACLR patients. In the introduction to the questionnaire participants are asked to imagine performing the movement depicted in the photograph and then they are asked to report the subjective FoH for each item on a scale from 0–10 (11-point rating scale). A score of 0 means “totally not damaging”, and 10 stands for “very damaging”. The average item score for the scale is computed. In a previous study, the PHOSA-ACLR showed excellent reliability and validity in patients with ACLR [20]. 

Sample 1 (ACLR patients) was asked to report time since reconstruction and completed two additional questionnaires assessing self-reported functioning and fear of movement. Self-reported functioning was assessed using the validated Dutch version of the Knee Injury and Osteoarthritis Outcome Score (KOOS) [22]. The KOOS measures pain, other disease-specific symptoms, Activities of Daily Life (ADL) function, sport and recreation function, and knee-related Quality of Life (QOL). For each scale, the scores were recoded from 0 to 100, with 100 depicting no problems. Fear of movement was assessed using the Dutch version of the Tampa Scale of Kinesiophobia (TSK) [18,23,24]. The TSK measures generic fear of movement or (re)injury using 17 items. Item scores range from 1 to 4, where 1 = strongly disagree and 4 = strongly agree. Total score is the count of all 17 items after inversion of items 4, 8, 12 and 16. A score higher than 36 is considered kinesiophobia. 

Healthy subjects in sample 2 and 3 were asked to complete two additional questions related to sports activities (which sports activities they were active in, and hours of sports per week), and three yes/no questions relate to physical condition (do you have a current medical condition, have you ever had previous lower extremity injuries, and do you know anyone with an ACL rupture).

### 2.3. Analysis

*Descriptive statistics* are given for the participants in the three groups. For continuous variables, the mean score is reported, standard deviation, and 95% Confidence Interval for Means (95% CI) when appropriate. For dichotomous variables, percentages are given for each group. For sample 1, ACLR-related mean scores are given for functioning (KOOS), and Fear of Movement (TSK). In sample 2 and 3, reported sports activities were recoded into two categories: low risk ACL tear sports (code 1 = swimming, horseback riding, ballet, running, etc.); high-risk ACL tear sports (code 2 = soccer, hockey, tennis, baseball, etc.). 

*Item Response Theory (IRT)* models are used to determine the probability of an individual’s response to an item, given their latent ability, which in this case would be ACLR-specific FoH [24]. In this study we applied a polytomous IRT model called a Graded Response Model (GRM) to the PHOSA-ACLR, in which the item parameters were estimated for each 11-point Guttman item used in this study. The GRM is an especially useful item response model when item response options can be conceptualized as ordered categories with a strongly restricted monotonicity, such as Guttman scales. Item locations are the polytomous equivalent to item difficulties in IRT models for dichotomous responses. A GRM is similar to IRT models with dichotomous outcomes, but within the GRM framework an item response scale reconstructs a polytomous scale in a series of *m-*1 dummy response dichotomies, where m represents the number of response options for a given item. Thus, an item rated on a 11-point scale has ten response dichotomies: category 1 versus category 2, category 2 versus category 3, etc. The graded response model estimates the probability of endorsing a response category for each item conditional on the latent trait, in which the latent trait is considered a weighted and standardized total scale score. Two models were tested: The first model was a 1-parameter (1-PL) model, in which only the item locations were estimated, and discrimination was set to one for all items (often referred to as a Rasch model). The second model that was tested was a 2-parameter model in which, in addition to the estimation of item locations, the item-discrimination parameters were estimated. Item discrimination equates to the slope of the item response function and is a measure of how sharply the response options discriminate between each other around their location parameters. Higher discrimination means that the items are more informative. To determine whether the added complexity of the 2PL model is necessary, the fit of the two models were compared on their -2loglikelihood, which is χ^2^ distributed. By testing whether the chi-square is significant, it was determined if the added complexity of the 2PL model significantly improved the model fit. 

*Differences between groups in PHOSA-ACLR.* Differences between groups were tested with Univariate Analysis of Covariance (ANCOVA). Before running an ANCOVA, possible covariates have to be identified. Covariates are variables that differ between groups under study and that are related to the dependent variables. Differences between samples in demographics were tested using χ^2^ for dichotomous variables, and ANOVA for continuous variables. Associations between demographic variables and PHOSA-ACLR scores were calculated using χ^2^ for dichotomous variables and Pearson correlation for continuous variables. Correlations were interpreted as small (r = 0.10), medium (r = 0.30), and large (r > 0.50) [25]. Normality of items distribution were tested using the Shapiro–Wilk test. Levene’s test of equality of error variance was used to test the hypothesis that the error variance of the dependent variable (PHOSA-ACLR) is equal across groups. F statistics and significance level are given for the corrected model comparing the three groups of participants controlling for covariates (ANCOVA). The same statistics are given for the independent variable (groups to be compared), and covariates. For significant co-variates, partial eta square was computed to determine the percentage of variance attributed to that covariate. All analyses were performed using SPSS version 26 (IBM Corporation). A *p*-value < 0.05 was used as an indication of statistical significance. 

## 3. Results

### 3.1. This Description of the Samples

Sample 1 included 58 ACLR patients (mean age 25.9 years; range 17–56; SD = 8.2; 43% male). Average time since reconstruction was 15.5 months (range 3–36; SD = 8.0). In Table 1, mean levels of self-reported functioning (KOOS) and fear of movement (TSK) are given.

Table 1 shows that patients in the ACLR group, on average, scored highest on the KOOS scales for Pain and ADL as compared to the other KOOS scales, indicating little pain, and high levels of ADL functioning. KOOS QOL showed relatively lower scores, indicating that quality of life was limited. Based on the TSK score, kinesiophobia (TSK score > 36 points) was present in 26 patients, or 45% of the patient sample.

Table 2 shows descriptive statistics of both samples of healthy participants. For continuous variables, mean and standard deviation (SD) are given. Sports activities reported by students were recoded into high risk and low risk sports activities. 

Sample 2 included 169 regular students (mean age = 19.2; SD = 2.0; 48% male). Students participated frequently in sports activities. Only six students (3.4%) reported that they did not actively participate in sports activities. On average, PT students reported to be active in sports for 5.6 h a week. Sample 3 comprised 30 soccer players from Vitesse (mean age = 18.3; range = 17–20; SD = 3.2; 94% males). This group reported an average of 14.9 h of active sporting per week. The three groups of participants differed in average age (F(2, 254) = 51.2, *p* < 0.001), with ACLR patients being of higher age compared to the student groups. The three groups of participants differed in gender (χ^2^ = 21.4, *p* < 0.001). The group of junior soccer players was predominantly male, with only two females. Group 2 differed in percentage high/low risk sports compared to both other groups, where all participants in group 1 and 3 were considered high-risk athletes.

### 3.2. Assessment of Graded Response Model

Prior to analysis, the PHOSA-ACLR items were examined for accuracy of data entry, and missing values. No uni- or multivariate outliers were found. The dimensionality of the PHOSA was tested with Principal Axis Factoring, which showed a clear unidimensional fit for the PHOSA items (TLI = 0.90, RMSR = 0.05) with factor loadings between 0.69 and 0.89. Descriptive statistics for the PHOSA items are summarized in Table 3.

The graded response model was used to calibrate item parameter estimates for the PHOSA given its Guttman-scaling; i.e., ordered polytomous response format. The best fitting GRM was first assessed by comparing two models, (1) a one-parameter model (Rasch) with location as parameter (and discrimination assumed fixed to 1); (2) a two-parameter model, with both location and discrimination as parameters. The two-parameter model (−2LL = −5485.05) showed a significantly better fit than the one-parameter model (−2LL = −5515.36),
$\chi^{2}$(1) = 30.31, *p* = 0.00). Subsequent analyses pertain to the fit of the two-parameter model in which both location and discrimination are estimated. In Table 4, the item difficulty locations are sorted, from top to bottom, in ascending order of increasing difficulty. 

How these locations relate to the thresholds within each item are summarized in Item Information Curves in Figure 1. 

The ICCs in Figure 1 indicate that items 8 and 10 are most informative in the range of −2 and 2, whereas item 12 is most informative below −2, and item 9 above +3. Item 6 is least informative in general. The thresholds in Table 2 correspond to the intersection between curves. At these intersections, it is equally likely a person will be classified into adjacent categories, and therefore to obtain one of two successive scores on the item. The mean of the threshold locations within an item is represented by the location parameter. The locations range from 1.69 to 3.48 logits, where “Hopscotch” (PHOSA_8) was the most difficult item and “Sliding” (PHOSA_6) was the easiest item (Table 4). 

In Figure 2, the distributions of ability estimates in the sample are summarized in a kernel density plot showing an approximately normal distribution that is slightly negatively skewed. 

The kernel density plot shows that the tails of the PHOSA distribution are not symmetrical and has a fat bottom tail. PHOSA scores 2SD or more above the mean are as uncommon as expected from a standard normal distribution, whereas PHOSA scores 2SD or lower are more prevalent than expected.

The Test Item Information Function, visualized in Figure 3 shows that the test is most informative (has its peak-information) for people within a standardized PHOSA-range of −2 to +2 standard deviations, with an evenly distributed peak in that range, indicating that the test is consistently reliable in this entire range.

### 3.3. Difference between Groups in FoH Assessed with the PHOSA-ACLR

Levene’s test of variance was significant (F (2, 253) = 5.1); *p* = 0.007), indicating that the error variance is not equal across groups. The one-sample Shapiro–Wilk was significant, indicating that the distribution of PHOSA-ACLR scores was not normal (Shapiro–Wilk statistic = 0.94, df = 257, *p* < 0.01). Table 5 depicts average PHOSA-ACLR scores for patients of each of the three samples, without any correction of covariates. 

On average, regular students reported the highest level of FoH assessed using the PHOSA-ACLR comparable with FoH reported in soccer players. Regular students, on average, reported higher levels of FoH compared to ACLR patients and soccer players. Within the healthy samples (samples 2 and 3), students participating in high risk sports did not differ from students with low risk sports activities in average PHOSA-ACLR scores; mean PHOSA-ACLR score in the high risk group is 2.06 (SD 0.22), compared to 2.4 (SD = 0.23) in the low risk group (F (2, 190) = 4.47, *p* = ns). A chronic health condition was reported by 17 students, and these students did not report different PHOSA-ACLR scores when compared to students without chronic condition. Over one third of the students (*n* = 73; 38%) reported to know someone with an ACL rupture. These students did not differ in mean PHOSA-ACLR from students who did not know anyone with an ACL rupture. Number of hours spent on sports on average per week showed weak association with PHOSA-ACLR (*n* = 198; r = −0.30, *p* < 0.001). Students spending more time on sports showed, on average, lower scores on PHOSA-ACLR. 

For all three samples, gender, age and previous injury are known and analyzed as possible covariates. The average PHOSA-ACLR score for males is 3.8 (SD = 2.3) compared to 4.7 (SD = 2.2) in females (t(254) = 2.98, *p* < 0.0001). The age of participants showed a small but significantly reversed correlation with PHOSA-ACLR (r = −0.21, *p* < 0.001). Previous injury at the knee was reported by 194 participants (including all participants from sample 1). Mean PHOSA-ACLR for these participants was 4.32 (SD = 2.3) and was not significantly different from the average score of 4.0 (SD = 2.6) in those participants not reporting previous injury (t(254) = 0.97, *p* = 0.35). Therefore, only gender and age are considered as covariates in the ANCOVA for differences between the three groups, as these are related to PHOSA-ACLR, and differ between groups. In the ANCOVA analysis, the covariate was unrelated to PHOSA-ACLR, with gender being the sole covariate with a significant relation to PHOSA-ACLR (F (1, 255) = 4.01; *p* = 0.04). Gender explained 1.6% of the variance in PHOSA-ACLR independently from the other variables in the equation (partial eta squared = 0.016). There was a significant difference between the three groups in PHOSA-ACLR after controlling for age and gender (F (2, 255) = 17.1, *p* < 0.001). Differences in groups explained 12% of variation in PHOSA-ACLR (partial eta squared = 0.12). Post-hoc pairwise comparison adjusted for multiple comparisons (Sidak) showed that the students differed from both the ACLR group (lower FoH, mean difference = 1.6, *p* < 0.001), and from soccer players (higher FoH, mean difference = 2.0, *p* < 0.001). As the distribution of scores on the PHOSA-ACLR was shown to be not normal, the ANCOVA was repeated using log transformation of PHOSA-ACLR (Log (PHOSA-ACLR+1). This analysis using transformed PHOSA-ACLR scores resulted in similar effect for group difference (F (2, 255) = 18,2, *p* < 0.001; partial eta squared for groups = 0.13).

## 4. Discussion

This study showed that the distribution of scores on the individual items of the PHOSA-ACLR fits the Item Response Model. It is therefore safe to assume that the individual items of the PHOSA-ACLR measure one underlying latent trait. As the PHOSA-ACLR items measure one latent trait, the items can be used in selecting items when treating fear of harm in sports-related activities. Contrary to expectation, the average ACLR patients in this study did not score higher on FoH assessed using the PHOSA-ACLR when compared to individuals without ACLR. 

The finding that ACLR patients on average scored lower on the PHOSA-ACLR than first year PT students is intriguing. Several theoretical explanations come to mind. First of all, it is possible that implicit negative beliefs related to the actions depicted in the PHOSA-ACLR exist in healthy individuals as well, and first-year PT students in particular, even without ever experiencing a negative effect of that action. This is consistent with the findings that people in the general population hold task-specific dysfunctional beliefs that are inconsistent with current evidence [26]. For instance, task-specific dysfunctional fear avoidance beliefs have been observed in healthy adults without back pain or prior back related aversive event [27,28]. As a result, people without pain display an implicit negative bias towards specific movements, such as bending and lifting, that are considered dangerous [29,30]. Similar processes might also be at work in PT students when asked to judge the harmfulness of actions depicted in the PHOSA-ACLR photographs. These high scores of starting PT students are particularly worrisome: health care providers fear avoidance beliefs and attitudes influence clinical practice [28,31,32]. Furthermore, fear avoidance beliefs of physical therapists might influence fear beliefs in their patients [33]. Although we did not assess FoH in practicing PT, these results indicate that it is important that PT students learn about dysfunctional negative biases related to sports-specific activities. PT education curricula should address such negative beliefs in students in an effort to correct them. Cultural differences offer an alternative explanation for the observed differences in fear-avoidance [34]. These differences in task-specific FoH reported in this study might reflect differences in subcultures in which the participants are involved, resulting in differences in community beliefs communicated to the respondents [26]. ACLR athletes, aspiring professional athletes and amateur athletes have different social networks or subcultures. Such differences in social network, including spouses, friends, but also trainers, physical therapists, etc., might account for differences in task-specific FoH between groups. Finally, in patients with an ACL, reconstruction exposure might be at work. Intensive exercises and training are needed to enhance the athlete’s performance in sports activities [2]. Therefore, it is likely that patients are frequently exposed to the sports-related activities assessed with the PHOSA-ACLR as part of their rehabilitation, diminishing levels of anxiety. The weak reversed relation between hours spent on sporting and PHOSA-ACLR score in the athletes without ACLR suggests that FoH for a specific situation diminishes when people are frequently exposed to that situation. More research is needed to determine how these interacting mechanisms are at work in athletes with and without ACLR. 

This study is not without its limitations. First of all, patients with ACLR were recruited at different times after the ACLR, and this might have an impact on the average score reported in this sample. However, PHOSA-ACLR was assessed between 3–36 months after the intervention, at which time most patients have recovered to normal use. Indeed, patients in this sample reported, on average, little pain, and high levels of ADL functioning. Furthermore, time since ACLR in this sample is unrelated to PHOSA-ACLR score [20]. The average scores on PHOSA-ACLR are lower compared to the average scores in the students, but higher in junior professional soccer players. A second limitation of the study is the sample of students starting with the study of PT. These students might have been primed on possible negative consequences of certain behaviors. However, to rule out such a bias as much as possible, the questionnaire was completed by the students at the start of the first college on an unrelated topic. As PT students are known to be frequent sporters, they were considered to be a good control group. However, to rule out any bias, more research is needed including samples of healthy subjects with no previous concerns about sports injuries. Furthermore, in this study, the professional soccer players were mostly male, and gender is an independent predictor of PHOSA-ACLR. Although this was controlled for in the analysis, it would be worthwhile to study FoH in an all-female professional soccer team. A final restriction of the study is the different ways the PHOSA-ACRL was assessed. In two samples, this was carried out on paper, in one sample, online. Based on previous studies with the PHOSA-ACLR, there is no indication that these differences in assessment resulted in different scores. Nonetheless, this possible bias cannot be entirely ruled out. 

Nevertheless, the study showed that task-specific FoH assessed with the PHOSA-ACLR is a unidimensional trait. FoH is a psychological barrier to returning to sports, particularly with ACL injuries where (re)injury is very commonly observed [35]. Therefore, clinical practitioners should be aware of and monitor such fear of (re)injury or FoH in ACL patients. Identifying at-risk patients with high levels of FoH, encouraging a multidisciplinary approach to patient care, and providing early referral to a sports psychologist may improve patient outcomes and increase return-to-play rates among athletes [15]. In patients with high levels of FoH, exposure cognitive behavioral therapy can be applied. Exposure involves disputing dysfunctional thoughts and images by systematically questioning the reality of the thought and images, starting with the least fear inducing situation in the hierarchy. Such engagement can be either real or imagined. Imagery exposure (motor imagery) can potentially reduce the fear of (re)injury and pain in ACLR patients [36].

Further research is needed to determine at what moment in the rehabilitation process FoH should be assessed. FoH is likely to be high shortly after the intervention but will decrease over time. However, longitudinal research is needed to determine at what point in time individual patients develop dysfunctional FoH, that is, FoH that will prevent the return to sport. It is important to determine the earliest point of time in rehabilitation when dysfunctional FoH can be determined. Initiating exposure-based treatment at that point in time is likely to be the most beneficial. Finally, it would be worthwhile to educate PT students and PT in general about the existence of implicit negative bias towards specific movements. Educating the PT might prevent dysfunctional fear beliefs in their patients.

## 5. Conclusions

Specific sports-related FoH assessed with the PHOSA-ACLR is a unidimensional trait. In clinical practice, the PHOSA-ACLR can be used to assess levels of sports-related FoH in patients with ACLR, and physical therapists can use the PHOSA-ACLR to guide exposure treatment.

## Figures and Tables

**Figure 1 ijerph-17-06764-f001:**
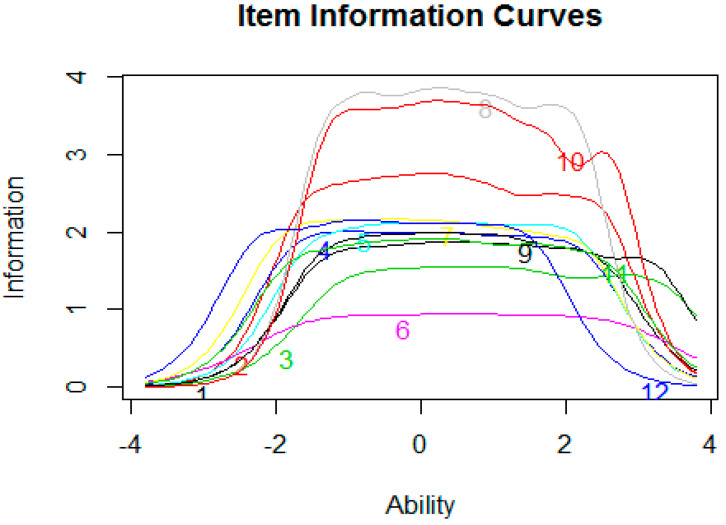
Item Information Curves (IIC) for the 12-item PHOSA. The amount of item-information (y-axis) is plotted against latent ability scores (x-axis). The amount of item-information (y-axis) is plotted against latent ability scores (x-axis). Each colored line represents an item, and how much information a response on that item provides at increasing standardized levels of ability (theta). For example, item 6 carries relatively low information across all ability levels, whereas answer items 8 and 10 are highly informative in the −2 to +3 ability range.

**Figure 2 ijerph-17-06764-f002:**
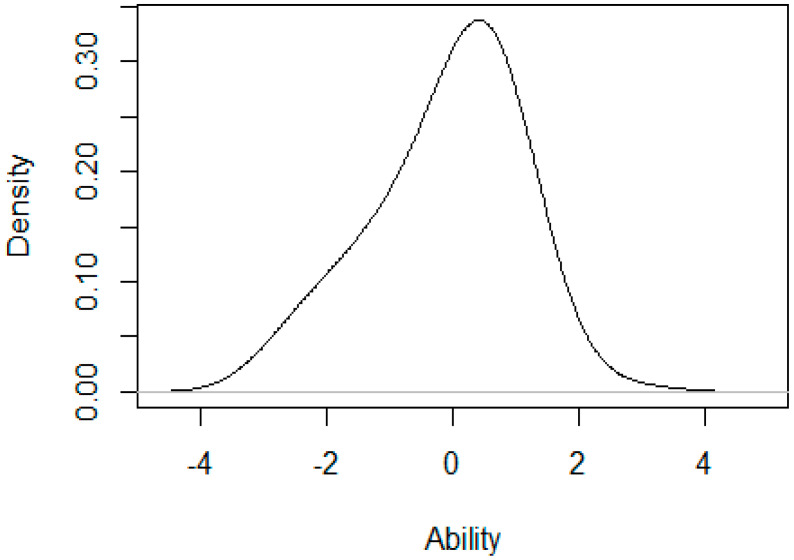
Density plot of the 12-item PHOSA visualizing the distribution of latent PHOSA scores (x-axis).

**Figure 3 ijerph-17-06764-f003:**
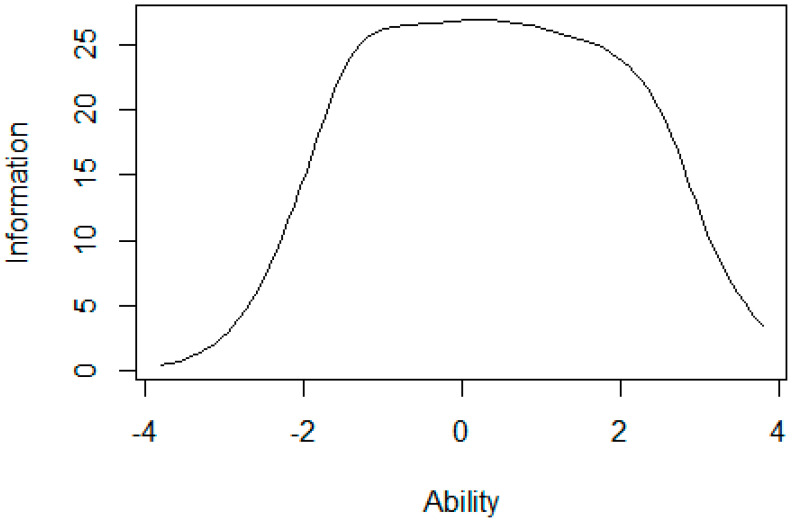
A visualization of the Test Information Function for the 12-item PHOSA over the range of ability scores (x-axis).

**Table 1 ijerph-17-06764-t001:** Characteristics of patients in rehabilitation after reconstruction surgery (*n* = 58).

Variable	Theoretical Range	Mean (Standard Deviation)	Observed Range
KOOS Pain	0–100	81.8 (17.7)	33–100
KOOS Symptoms	0–100	63.2 (12.8)	29–100
KOOS Activities of Daily Life	0–100	89.1 (14.2)	35–100
KOOS Sports and Leisure	0–100	59.1 (30.6)	00–100
KOOS Quality of Life	0–100	48.3 (14.1)	06–75
Tampa Scale of Kinesiophobia	0–68	35.7 (7.1)	20–51

Abbreviations: *n* = number of participants, KOOS = Knee Injury and Osteoarthritis Outcome Score.

**Table 2 ijerph-17-06764-t002:** Description of student samples (*n* = 199).

Variable	Regular Students *n* = 169	Vitesse *n* = 30
Average age (mean, SD)	19.2 (2.0)	18.3 (3.2)
Average hours active sporting (mean, SD)	5.6 (2.8)	14.9 (4.2)
Gender (% male)	48%	94%
Earlier complaints (% yes)	71%	60%
Knowledge of ACLR (% yes)	35%	65%
Chronic condition (% yes)	10%	7%
High risk sports (% yes)	66%	100%

Abbreviations: SD = Standard deviation; ACLR = Anterior Cruciate Ligament rupture.

**Table 3 ijerph-17-06764-t003:** Photographic Sports Activity (PHOSA) items, observed range, means, standard deviations, and median score.

Sports Related Activity	Observed Range	Mean Score (SD)	Median
1. Running	0–10	3.67 (2.73)	3.0
2. Landing after jumping	0–10	4.39 (2.90)	4.0
3. Squats	0–10	3.11 (2.63)	3.0
4. Lateral lunging	0–10	4.76 (2.83)	5.0
5. Single leg jump	0–10	3.88 (2.71)	4.0
6. Sliding	0–10	3.98 (2.79)	4.0
7. Bring to a halt	0–10	5.11 (2.85)	6.0
8. Hopscotch	0–10	4.11 (2.84)	4.0
9. Lunge	0–10	3.77 (2.78)	4.0
10. Start to sprint	0–10	4.09 (2.84)	4.0
11. Jumping on a trampoline	0–10	4.06 (2.77)	4.0
12. Pivoting movement	0–10	5.68 (2.86)	6.0

**Table 4 ijerph-17-06764-t004:** Location parameters of the PHOSA items, sorted from lowest difficulty to highest.

PHOSA Item	β.1	β.2	β.3	β.4	β.5	β.6	β.7	β.8	β.9	β.10	β
6. Sliding	−2.73	−2.03	−1.10	−0.20	0.43	1.15	1.80	2.64	3.50	4.73	1.69
3. Squats	−2.42	−1.15	−0.19	0.80	1.59	2.30	2.70	3.75	5.82	7.25	2.18
1. Running	−3.30	−2.04	−0.63	0.30	0.86	1.79	2.73	3.85	5.39	5.96	2.40
11. Jumping	−4.31	−2.56	−1.32	−0.35	0.49	1.24	1.92	3.52	4.42	6.19	2.42
9. Lunge	−3.40	−2.03	−0.97	−0.07	0.70	1.63	2.53	3.47	5.34	7.82	2.47
4. Lateral Lunging	−4.48	−3.45	−2.49	−1.60	−0.40	0.65	1.36	2.76	4.12	5.67	2.49
5. Leg jump	−3.95	−2.33	−1.33	−0.28	0.61	1.72	2.58	3.89	4.73	5.76	2.56
12. Pivoting	−5.92	−5.32	−3.56	−2.59	−1.91	−1.03	0.37	1.44	2.80	4.03	2.58
7. Bring to Halt	−5.08	−3.84	−3.01	−1.79	−0.98	−0.08	1.00	2.14	3.85	5.84	2.59
2. Landing	−4.73	−3.12	−1.74	−0.69	0.31	1.02	1.67	3.14	5.23	7.28	2.92
10. Start sprint	−4.46	−3.28	−1.81	−0.40	0.63	1.49	2.77	3.92	5.88	8.81	3.39
8. Hop	−4.69	−3.09	−2.11	−0.53	0.68	1.43	2.78	4.12	6.07	7.67	3.48

β.1 = Beta.1 represents the threshold between response category 0 and 1, Beta.2 represents the threshold between response category 1 and 2, etc. The final Beta indicates the item’s overall difficulty—the ability (theta) associated with the weighted threshold between response category 0 and 10.

**Table 5 ijerph-17-06764-t005:** Average PHOSA score, range, Standard Deviation (SD) and 95% Confidence Interval for patients after ACLR, regular students, and soccer players.

	ACLR *n* = 58	Regular Students *n* = 169	Soccer *n* = 30	Total *n* = 257
Average PHOSA	3.1	4.9	2.6	4.2
Range	0–10	0–10	0–10	0–10
Standard Deviation (SD)	2.3	2.0	2.7	2.3
95% CI	2.5–3.7	4.6–5.2	1.6–3.6	3.9–4.5
Median	2.5	5.5	1.6	4.8

## Data Availability

The datasets generated and/or analyzed during the current study are available in the DANS KNAW repository, 10.17026/dans-zdq-tyjx.

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
