# Peer review of "Is Fear of Harm (FoH) in Sports-Related Activities a Latent Trait? The Item Response Model Applied to the Photographic Series of Sports Activities for Anterior Cruciate Ligament Rupture (PHOSA-ACLR)"

_ijerph, 2020, doi:10.3390/ijerph17186764_

Round 1

Reviewer 1 Report

Although the authors defended/revised their previous submission, the resubmission remains major statistical concerns that lead to question about the power and reliability of measurements. Despite the authors insisted the strength of ANCOVA analysis in comparison three unequal number of groups, they missed an essential stage to rectify the raw data prior to parametric measures. Such a large difference among the groups (ACL patients = 58, university students = 169; football players = 30) would decrease statistical power and increase type I error, affecting assumption of equal variance in ANCOVA test (Rusticus & Lovato, 2014. Practical Assessment, Rsearch & Evaluation).

Furthermore, the authors failed to provide decent responses to assert absence of correlation and ICC outputs in the results section. When you mentioned the statistical methods in the Materials and Methods section, you cannot neglect the data for presentation.

Reviewer 2 Report

.

Reviewer 3 Report

The authors have addressed all my concerns

This manuscript is a resubmission of an earlier submission. The following is a list of the peer review reports and author responses from that submission.

Round 1

Reviewer 1 Report

After reading this paper, in my points of view, there are several major methodological and analytic issues in this study. First of all, the sample size estimation is questioned due to sample size difference and gender difference among the group. Second, one group using online questionnaire and another two groups using hard copy of questionnaire, I think this is the major issue to limit the study. Thirdly, the authors should focus on statistical analyses in ANCOVA and graded response model (the normality of the data). It is unclear why the authors used correlation and ICC to compare the data, even though no reports in the results section. Lastly, the outcome of the study is controversial as the normal participants reported higher hear of harm. I hope the comments provided are useful.

  1. Participants’ profile should be reported in the abstract.
  2. Statistical report should be mentioned in the abstract.
  3. It is inappropriate that the conclusion just simply descript the result of the study.
  4. Line 39, the sentence is unclear.
  5. The rationale to conduct the fear of harm in three different sample pool is missing in the introduction. In addition, I do not think the introduction leading a flow to the purpose of the study.
  6. How the authors calculated 95% CI. It is interesting to me that only one number presented for 95% CI.
  7. The presentation of tables can be improved. It is strange that standard deviation and median are used in the same table.
  8. The figures are difficult to read.
  9. The discussion is too board to read as the authors failed to discuss the major findings and comparable results to previous studies.

Reviewer 2 Report

Dear authors, thank you for the work you have presented. I consider that it has great quality.

RESULTS

L: 239 This statement should be an error. Do you agree? Please, should be corrected.

'The kernel density plot shows that relatively high PHOSA-scores are rare'

Reviewer 3 Report

Major comments

Introduction- It was a very thorough introduction. Except for a few minor errors I feel that it did a good job of making a case for why the study should be conducted.

Methodology: 

In the participants section please provide the fact that there was ethics approval. Were the three samples from 3 different studies? If so please acknowledge in the participants section that all 3 studies had ethics approval. 

You mention checking for normality. Were all variables normally distributed? If not, what was done?

Results

The 6 students who reported not being involved in sports, did you take them out of your analysis? Were the results the same?

For the analyses between high risk sports and low risk sports, did you compare just between students?

What transformation techniques did you use to transform the PHOSA-ACLR? Please attach those results in a supplemental table. 

Overall you did a great job with the statistical analyses and results. 

Discussion

I believe you should point out in your discussion that these were first year PT students. Although I am unaware of evidence for physical therapy students, there is evidence that after students have had certain modules of schooling their beliefs change (Fitzgerald, K., Fleischmann, M., Vaughan, B., de Waal, K., Slater, S., & Harbis, J. (2018). Changes in pain knowledge, attitudes and beliefs of osteopathy students after completing a clinically focused pain education module. Chiropractic & manual therapies, 26(1), 42.). I don't know if this particular population had already completed the orthopedic module of their program, but I would assume that after the orthopedic module their FoH scores might change. 

I do not think that using attitudes of first year PT students to extrapolate the they might take this attitude into their clinical practice is warranted. Perhaps you can state that this may carry over into their clinical practice however, you didn't test students who were getting ready to enter clinical practice so you don't know whether the students attitudes changed after specific course modules.

I know you address this in your limitations section, but I think making such a strong statement beforehand is not warranted.

In your limitations section you need to address that perhaps these attitudes might change as these students get closer to graduation. 

Minor comments

In the title, please capitalize "the" after the colon

The second paragraph of the introduction, although well written is entirely too long. It can be broken up into 2 paragraphs. One outline the use of the PHOSA-ACLR, the other about the construct validity and why it's important to explore those who are low and normal trait. (probably around the sentence "Although strongly associated..."

Methods

Define HAN before abbreviating.

Results

Line 197- risk was misspelled